# Mitomycin C in Homologous Recombination Deficient Metastatic Pancreatic Cancer after Disease Progression on Platinum-Based Chemotherapy and Olaparib

**DOI:** 10.3390/biomedicines10112705

**Published:** 2022-10-26

**Authors:** Gehan Botrus, Denise Roe, Gayle S. Jameson, Pedro Luiz Serrano Uson Junior, Ronald Lee Korn, Lana Caldwell, Taylor Bargenquast, Max Miller, Erkut Hasan Borazanci

**Affiliations:** 1HonorHealth Research Institute, Scottsdale, AZ 85258, USA; 2Winship Cancer Institute, Emory University, Atlanta, GA 30322, USA; 3UA Cancer Center, University of Arizona, Tucson, AZ 85719, USA; 4Center for Personalized Medicine, Hospital Israelita Albert Einstein, São Paulo 05652-900, Brazil; 5Scottsdale Medical Imaging, Ltd., Scottsdale, AZ 85258, USA

**Keywords:** pancreatic cancer, BRCA, Olaparib, homologous recombination deficient, mitomycin

## Abstract

Recent efforts to personalize treatment with platinum-based chemotherapy and PARP inhibitors have produced promising results in homologous recombinant deficient (HRD) metastatic pancreatic cancer (MPC). However, new strategies are necessary to overcome resistance. The below case series documents patients treated at the HonorHealth Research Institute with a diagnosis of HRD MPC who received Mitomycin C (MMC) treatment from January 2013 until July 2018. Five HRD MPC patients treated with MMC were evaluated. All patients received at least one course of treatment. Mean age at MMC treatment initiation was 58 years. There were 3 females and 2 males. All patients had tumors that progressed on platinum-based chemotherapy, four patients had previous exposure to Olaparib. The median PFS was 10.1 months, and the median OS was 12.3 months. Responses were observed only in patients harboring *BRCA2* mutations, no response was observed in the *PALB2* mutation carrier. MMC in this heavily previously treated PC was safe, with overall manageable grade 2 gastrointestinal toxicities including nausea and vomiting, and G3 hematological toxicities including anemia and thrombocytopenia. Pancreatic cancer patients with HRD may benefit from MMC treatment. Further clinical investigation of MMC in pancreatic cancer is warranted.

## 1. Introduction

Pancreatic cancer (PC) continues to be a leading cause of cancer related deaths in the United States and worldwide. In the United States alone, PC accounts for 3% of all cancers and 7% of all cancer related deaths, with an estimated 48,830 deaths in 2022 [1]. Proportionally, men are at a slightly higher risk for this disease than women [1].

Given the vast majority of pancreatic ductal adenocarcinoma (PDAC) patients present with locally advanced or distant metastatic disease, the overall 5-year survival rate is 11%, a significant increase from only 7.8% in 2008 [1,2,3]. Currently, surgical resection, if the disease is detected early, remains the only potential cure for pancreatic cancer. Only approximately 15–20% of PDAC patients are candidates for potentially curative resection, and unfortunately as high as 80% of these patients see recurrence within 2 years of resection for which the prognosis is poor [4,5].

Approximately 5–10% of PC patients have a family history of the disease, indicating a genetic basis for their susceptibility [6]. Given the limited treatment options, recent efforts have been made to personalize treatment with genetic profiling and have produced promising results. For instance, agents that suppress DNA repair mechanisms such as Poly(ADP-ribose) polymerase inhibitors (PARPi), which inhibit the PARP enzyme and trap DNA-PARP complexes, offer promise in not only inducing DNA damage but in increasing the efficacy of current chemotherapies [7,8]. Tumors with *BRCA1* and *BRCA2* mutations have demonstrated increased susceptibility to PARP inhibitors given the tumor cells defects in chromosomal repair and error-free DNA double-stranded breaks [9]. In 2019, PARPis were granted FDA approval for maintenance therapy in germline *BRCA1/2* PC [10,11]. Mutations in *PALB2*, which encodes proteins necessary for the BRCA2-PALB2-Fanconi DNA repair pathway, and *BRCA2* have also been implicated as a risk factor for PC [12,13]. With this, genomic analyses have uncovered new therapeutic targets that may inform personalized cancer therapy [13,14]. In addition, patients with pancreatic tumors that harbor homologous recombinant deficient (HRD) phenotypes have better treatment results with either alkylating agents including platinum-based chemotherapy or rucaparib, another PARP inhibitor [15,16].

In addition to PARP inhibitors, cancers with biallelic inactivation of the genes encoding *BRCA2* and *PALB2* may be particularly sensitive to another alkylating agent, mitomycin C (MMC) [17]. Mitomycin C is a naturally occurring quinone that acts as an alkylating agent and DNA crosslinker, inhibiting the transcription of DNA to RNA thus inhibiting protein synthesis [18,19]. MMC has demonstrated antitumor efficacy against non-small cell lung cancer and tumors including gastrointestinal, head and neck, esophageal, and bladder [19]. In one case report, a patient with biallelic inactivation of *PALB2* exhibited resistance to gemcitabine. After treatment with MMC, CA19-9 levels were markedly reduced to normal levels after 3 years and the patient remained asymptomatic, suggesting a potential therapeutic effect of MMC for individuals with PC with genomic changes consistent with DNA repair abnormalities [13]. Murine models of gastrointestinal malignances with BRCA mutations were also consistent with these findings, indicating that MMC could be an option for gastrointestinal tumors with BRCA deficiency [20].

This retrospective study investigates the impact of MMC in individuals with PC who have pathogenic germline variants (PGV)s in either *BRCA1/2* or *PALB2* and were treated at a single institution.

## 2. Materials and Methods

### 2.1. Study Population

We performed a retrospective chart review from records at HonorHealth Research Institute (HHRI) to identify patients with a diagnosis of PC who received MMC treatment from January 2013 until July 2018.

### 2.2. Data Collection

After HonorHealth IRB approval, data were obtained on individuals’ demographics, tumor marker levels, treatment history, germline and somatic sequencing results, and personal and family history of cancer. Clinically significant adverse events were graded according to CTCAE v 5.0 (Washington, DC, USA) [21].

### 2.3. Genetic Information

All germline testing was performed in Clinical Laboratory Improvement Amendments (CLIA) certified clinical diagnostic laboratories. Germline testing was performed through GeneDx (Gaithesberg, MD, USA), Invitae (San Francisco, CA, USA). Somatic testing, if done, was performed from tumor tissue through either Caris Life Sciences (Phoenix, AZ, USA) or Foundation Medicine (Cambridge, MA, USA) through CLIA certified clinical diagnostic laboratories.

### 2.4. Statistical Analysis

The primary endpoint was overall survival (OS), with secondary endpoints of progression-free survival (PFS), CA 19-9 dynamics, and tolerability.

## 3. Results

### 3.1. Demographics

A total of 5 patients with PC were treated with MMC and their demographics are summarized in Table 1. The median age when beginning treatment was 58.3 years (SD 16.5 years); patient ages ranged from 30 to 70 years. Three patients were female, and only 1 patient had a prior malignancy (breast cancer diagnosed at age 56). The average number of prior regiments prior to MMC therapy was 3.2. Four individuals displayed metastasis to the liver, and the remaining individual to the lung.

Table 2 displays each patient’s germline and somatic mutations along with the classification of pathogenic or variant of unknown significance (VUS). Family history of cancer was present in 4 out of 5 patients. The patient without a family history of cancer had several germline mutations: a pathogenic *PALB2* mutation, and a VUS in *ATM, MLH1, CDK4* and *BRCA1*. No patient had a family history consistent with familial pancreatic cancer.

Germline mutations consistent with Lynch syndrome were not found in any of the patients. All patients had homologous recombinant deficient (HRD) tumors. Germline mutations consistent with hereditary breast and ovarian cancer syndrome (HBOC), in *BRCA1* and *BRCA2*, were found in four patients. The most common germline mutation was in the *BRCA2* gene, in 3 out of 5 patients. The other mutations seen were pathogenic *PALB2* and *BRCA1* mutations, and VUS in *MLH1*, *ATM*, *CDK4*, *BRCA1*, *BRCA2*, *MSH6*, and *POLD1*. All patients were treated with an initial dose of MMC at 8 mg/m^2^ monthly. At the time of data cut-off, none of the patients are still on MMC and all patients have passed away. CA 19-9 dynamics can be seen on Figure 1.

### 3.2. Cases

#### 3.2.1. Case (1)

The first patient evaluated was a 69-year-old female with initial presentation of metastatic PC to the liver, positive for *BRCA2* gene. Microsatellite stable disease, CDKN2A/2B deletion, *BRCA2* E1035*, KRAS Q61R, TP53 S16* were all confirmed via somatic testing with a Tumor mutational burden (TMB) of 4. The patient was enrolled in a phase II study for approximately one year, consisting of Gemcitabine, nab-paclitaxel, and cisplatin. Patient began to develop a rash and exhibited pruritus on neck and hands around Cycle 9 of treatment. At cycle 13, patient experienced anaphylaxis after being admitted to the hospital and placed on a desensitization protocol designated for cisplatin. Cisplatin was discontinued at C13D8. Their best overall response was a partial response (PR) to gemcitabine, nab-paclitaxel, and cisplatin. A maintenance dose of Olaparib at 400 mg was then prescribed twice a day. After adverse side effects including nausea, vomiting and GERD despite a 50% reduction in dosage, Olaparib was discontinued. CA19-19 levels decreased from 475 U/mL to 15.4 U/mL while on Olaparib, however the disease had progressed with an enlargement of her pancreatic mass and a rise in CA19-9 again to 62 U/mL after ceasing treatment.

MMC was initiated at a dose of 8 mg/m^2^, and although MMC was discontinued after 3 cycles due to grade IV diarrhea and colitis, the patient exhibited SD. The patient had ceased treatment for 2 years and continued to exhibit controlled disease, one hepatic focal lesion had disappeared entirely, and two other hepatic lesions remained stable. Following these two years, her CA19-9 raised to 81 U/mL, and the patient presented with a new hepatic lesion, along with an increase in the size of her pancreas mass. MMC was initiated for a second time at 8 mg/m^2^ and continued for 3 cycles but was discontinued due to recurrence of high-grade diarrhea and thrombocytopenia. During treatment, CA19-9 levels again decreased from 81 U/mL to 28 U/mL. The patient held off on treatment for an additional 9 months with SD (Figure 2), after which she presented with an additional hepatic lesion, increased size of other hepatic lesions, and an increased CA19-9 level of 149 U/mL. Olaparib was started again at a lower dosage and later combined with bevacizumab to combat PD with rising CA19-9 levels and enlarged hepatic lesions. The patient was treated for two cycles after MMC was initiated a final time at 6 mg/m^2^, but discontinued due to PD. A final measurement of CA19-9 levels indicated an increase from 442 U/mL to 452 U/mL. Other short-term treatment options were explored prior to the patient’s passing.

#### 3.2.2. Case (2)

The second patient evaluated was a 61-year-old Caucasian female with initial presentation of hepatic metastasis from a primary PC. This patient was a germline carrier of the *BRCA2* gene and somatic testing was performed. This patient was determined to have microsatellite stable disease, MTAP loss exon 2-8, KRAS G12D, CDKN2A/B, *BRCA2* Y3092fs*11 mutation, ATM L1675, and SMAD4 Q250. The patient underwent 6 cycles of treatment on a Phase II study consisting of nab-paclitaxel, gemcitabine, and cisplatin and achieved a PR. After stereotactic body radiation therapy (SBRT) to the primary pancreas lesion, causing inflammation near the duodenum, no lesions were observed in the liver. CA19-9 levels showed improvement and decreased from 1086 U/mL to 17 U/mL. Patient was started on Olaparib and continued treatment for 10 months until a development of pneumonitis led to its discontinuation for 7 months. During the treatment break, CA19-9 levels increased to 269 U/mL, although scans indicated SD. When MMC treatment was re-initiated and shortly after, CA19-9 levels were measured at 466 U/mL and the pancreatic mass had increased in size. After 4 cycles of MMC, CA19-9 levels decreased to 73 U/mL and the patient exhibited SD (Figure 3). Due to an emergency, the patient ceased treatment. 5 months later, MMC was restarted for one cycle. Post cycle 1, the patient’s health declined rapidly and they were soon transferred to hospice, as the patient developed ascites and presented with biliary obstruction.

#### 3.2.3. Case (3)

The third patient evaluated was a 70-year-old Asian male with initial presentation of hepatic metastasis from a primary PC. This patient was a germline carrier of the *BRCA2* gene and somatic testing determined to have a PGV in *BRCA2*; *BRCA2* exon 11 p.K964N (VUS), MGMT, ERCC1, TUBB3, and TOPO1 positive by IHC, ARID1A and KRAS mutated, and exhibited microsatellite stable disease. Patient underwent 9 cycles on a Phase II study with cisplatin, gemcitabine, and nab-paclitaxel, and showed PR and CA19-9 levels decreased drastically to 120 m/L from 5543 U/mL. Maintenance therapy with Olaparib was started and CA19-9 continued to decline to 55.4 U/mL. Presentation of nodule in left lung was thought to be progressive disease (PD) and was later confirmed as primary lung adenocarcinoma. 5-FU plus liposomal irinotecan therapy was initiated, but intolerability led to its discontinuation after 5 months. Use of PARPi was discussed, however the patient elected to pursue treatment with MMC at 8 mg/m^2^ due to difficulties with insurance coverage with PARPi. After four cycles of MMC, two hepatic focal lesions were no longer present and a reduction in pancreatic mass size was observed. Despite these promising results, the patient’s CA19-9 levels increased from 72 U/mL to 231 U/mL and the left lung lesion had enlarged, and presented with increasing loculation and nodulation, therefore MMC treatment was discontinued (Figure 4). Mediastinoscopy was performed and determined to be negative for metastases through IHC and pathology, but rather a second primary lung adenocarcinoma. Patient initiated an alternative therapy with SBRT followed by a combination therapy consisting of Pemetrexed, Carboplatin, and Pembrolizumab but succumbed to the disease less than a year later.

#### 3.2.4. Case (4)

The fourth patient evaluated was a 58-year-old Caucasian female with initial presentation of hepatic metastasis from a primary PC and was *BRCA2* positive. Patient began treatment on a Phase II study (gemcitabine, nab-paclitaxel, and cisplatin) for 6 cycles which resulted in a decrease in CA19-9 levels from 1695 U/mL to 42 U/mL and a PR. Maintenance therapy with Olaparib was started and the patient’s CA 19-9 levels decreased to 25.4 U/mL, but was discontinued as anemia, renal insufficiency, and pneumonitis arose. PD led to a combination therapy of paricalcitol, gemcitabine, and cisplatin, though cisplatin was discontinued after 5 months due to nephrotoxicity. The gemcitabine and paricalcitol combination therapy was continued for an additional 4 months, during which marker levels increased to 346.7 U/mL, a new liver lesion arose, and the pancreatic mass had enlarged. Patient was treated with MMC at 8 mg/m^2^ in combination with paricalcitol for 4 cycles, during which hepatic lesions exhibited SD (Figure 5) and CA19-9 levels decreased to 106 U/mL. Treatment was later discontinued due to progressive disease, after which patient lost follow-up.

#### 3.2.5. Case (5)

The fifth patient evaluated was a 30-year-old Hispanic male with initial presentation of bone and lung metastases from a primary PC. This patient was a *PALB2* germline carrier and was determined via somatic testing to have to have microsatellite stable disease and mutations on ATM (P604S), TP53 (W146X), and *BRCA1* (V1804D) (VUS), with low expression by IHC of TOP2A, TLE3, TUBB3, and PGP. Patient had pericardial and pleural metastasis with effusion upon initial diagnostic visit. Patient began treatment with FOLFIRINOX and continued this regimen for 2 months during which the patient’s CA19-9 levels rose from 50.8 U/mL to 250 U/mL. Following this treatment, combination therapy of gemcitabine and nab-paclitaxel was given for a 5-month period. The patient’s disease had advanced, with development of ascites and lymphangitis carcinomatosis. One cycle of MMC was initiated. Patient complained of chest pain, a cardiac workup was performed and was unremarkable and negative for acute myocardial infarction though CA19-9 levels had risen again from 271 U/mL to 403.7 U/mL. Unfortunately, the patient was transferred to hospice care shortly after due to insurance coverage loss and rapidly deteriorating condition.

### 3.3. Outcomes

Five patients treated with MMC were evaluated (Table 3). Patient 1 was treated with three courses of Mitomycin C, whereas the other patients received one course. All analyses were based on patient 1′s first course. The mean age at Mitomycin C treatment initiation was 58.3 years (standard deviation = 16.5 years); patient ages ranged from 30 to 73 years. There were 3 females and 2 males. There were 3 Caucasians, 1 Asian and 1 Hispanic. Progression-free survival (PFS) is shown in Figure 6. Patient 3 did not progress and died of lung cancer, so was considered censored at the time of death. The median PFS was 10.1 months (95% CI = 0.9 months, upper bound not reached). Overall survival is shown in Figure 7. Patient 3 was again censored at the time of death. The median OS was 12.3 months (95% CI = 2.2 months, upper bound not reached) (Table 3).

### 3.4. Safety

Considering that most patients were treated with MMC beyond third line systemic treatment for metastatic disease, MMC treatment was fairly tolerated. The most common treatment related adverse events (TRAE) were gastrointestinal, including nausea, vomiting and diarrhea. One patient developed a grade 4 gastrointestinal TRAE due to colitis. Hematological toxicities were also common, including anemia and thrombocytopenia, observed in all patients (Table 4).

## 4. Discussion

This case series investigates the clinical outcome of MMC treatment on patients with germline *BRCA1/2* or *PALB2* metastatic PC. To our knowledge, this is the first case series detailing efficacy of MMC in germline positive HRD metastatic PC after failure with PARP inhibitors and platinum-based chemotherapy regimens.

Recent studies have implicated that PGV in genes such as *PALB2* and *BRCA2* in the DNA repair pathway in PC could be identified as a biomarker for anticancer therapy [22,23]. The *PALB2* gene product is a tumor suppressor and interacts with *BRCA1* and *BRCA2* to form a BRCA complex during double-strand break repair. The BRCA complex then interacts with RAD51 during homologous recombination. Homologous recombination is a process related to accurate DNA repair of DNA double-strand breaks, protecting the cells from aberrant defects [24]. It is estimated that around 15–20% of pancreatic cancer harbors some type of PGV, with the majority related to HRD genes [25,26]. HRD tumors are more responsive to platinum-based chemotherapy due to impaired DNA repair pathways [22,23]. Furthermore, these tumors demonstrate fragility to exposure to PARPis, with a synthetically lethal interaction due to accumulation of double-strand DNA breaks [13,20]. Nowadays, olaparib, a PARPi, is FDA approved for maintenance treatment of patients with deleterious germline *BRCA*-mutated metastatic PC, with controlled disease for at least 16 weeks of a first-line platinum-based chemotherapy regimen.

MMC may be especially effective in cells with mutations in the DNA repair machinery because the drug is a DNA-intercalating agent that crosslinks and damages DNA. Although alkylation of DNA is the most favored mechanism of action, other pathways such as inhibition of rRNA and protein synthesis also contribute to the drug’s action at higher dosages [27]. When mutations are present in *PALB2* or *BRCA1/2* gene, the DNA repair complex cannot work properly. Therefore, the damage on DNA from MMC leads to tumor cell death [27].

MMC has shown promising results in bladder, breast, anal, colorectal, and ovarian cancers [28,29,30,31]. One study by Moiseyenko et al. reported the clinical efficacy of MMC in 12 ovarian cancer patients with germline *BRCA1* mutations, suggesting that MMC may target defects in the DNA repair pathway, demonstrating increased activity in patients harboring HRD tumors [31]. Another study found that MMC in combination with methotrexate was effective for metastatic breast cancer patients who have had aggressive treatment with other therapies [32].

Pancreatic cancer patients have also benefited from MMC. Heinrich et al. reported the efficacy of gemcitabine combined with MMC in patients with advanced pancreatic cancer [33]. Thirty-seven treatment cycles were carried out in 17 patients. However, in this study, patients were treated with intra-arterial infusion into the celiac artery with MMC at 8.5 mg/m^2^ and gemcitabine 500 mg/m^2^ on days 1 and 22, with further intravenous infusions of gemcitabine 500 mg/m^2^ on days 8 and 15. The combination treatment resulted in tumor response in 24% of patients, with biomarker response in 41%. Median PFS for this study was 4.6 months and median OS of 9.1 months [33]. MMC has also been shown to be effective in a metastatic PC patient who had a *PALB2* pathogenic mutation [13]. After failure to gemcitabine and based on a xenograft model, the patient was treated with MMC 8 mg/m^2^ for 5 cycles [13]. The patient exhibited clinical response and dramatic lowering of CA 19-9 [13]. However, in our analysis, we did not see any significant benefit to the patient with *PALB2* mutation. This patient had overall survival of 2.2 months and CA 19-9 levels increased during treatment. Other groups have evaluated MMC in combination with other agents such as chemotherapy or olaparib in advanced pancreatic and biliary cancers, however those combinations did not reach safety to be evaluated in larger cohorts [34,35].

Four patients in this series had *BRCA2* PGV. In all cases, initially significant CA19-9 response was observed with MMC treatment despite previous exposure to platinum-based therapies and PARPi. CA19-9 is a biomarker related to progression of disease and poor outcomes in PC [36,37]. Studies with other agents, including gemcitabine and liposomal irinotecan, have associated clinical benefit of the treatment with decline of CA19-9, and this observation could be applied with MMC treatment [38,39,40]. However, larger cohorts would be necessary to confirm these findings.

Considering the poor outcomes of these patients with systemic treatment and that most of them were treated beyond second or third line systemic treatments, this case series provides rationale for further studies with mitomycin C and other agents related to cross-linking DNA damage beyond platinum-based therapies in this subgroup of patients, particularly *BRCA* carriers. Pancreatic cancer patients with *BRCA1/2* and *PALB2* mutations may benefit from MMC treatment. These results suggest that inactivation of genes in the DNA repair pathway provides a new approach for personalizing treatment. Further clinical investigation of MMC in pancreas cancer is warranted.

## Figures and Tables

**Figure 1 biomedicines-10-02705-f001:**
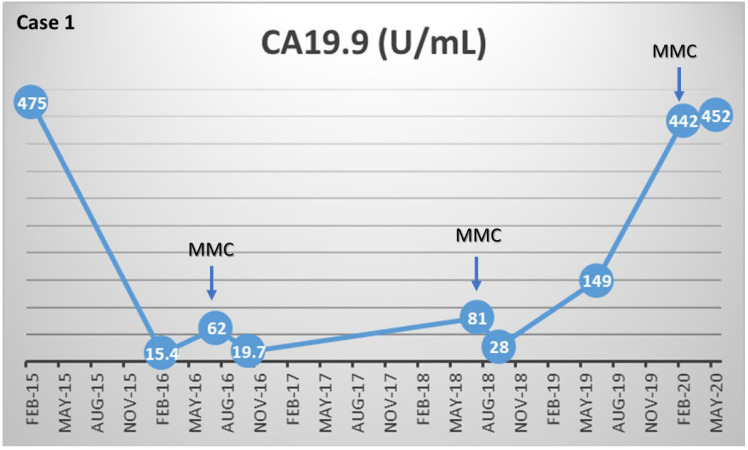
CA 19.9 dynamics on MMC treatment. Legend: Case 1: In two MMC exposures, reduction of CA 19-9 levels was observed. Case 2: MMC exposure resulted in an important decrease of CA 19-9 levels. Case 3: MMC did not resulted in decreased CA 19-9 levels. Case 4: After four cycles of MMC an expressive CA 19-9 decrease was observed. Case 5: Although MMC treatment, CA 19-9 levels raised.

**Figure 2 biomedicines-10-02705-f002:**
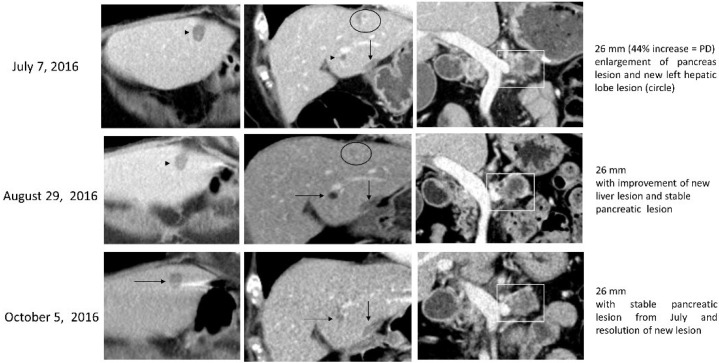
Case 1 treatment responses. Legend: On August 2016, during MMC treatment, we can see improvements in the liver lesion (arrow), furthermore in October of 2016, resolution of another liver lesion (circle).

**Figure 3 biomedicines-10-02705-f003:**
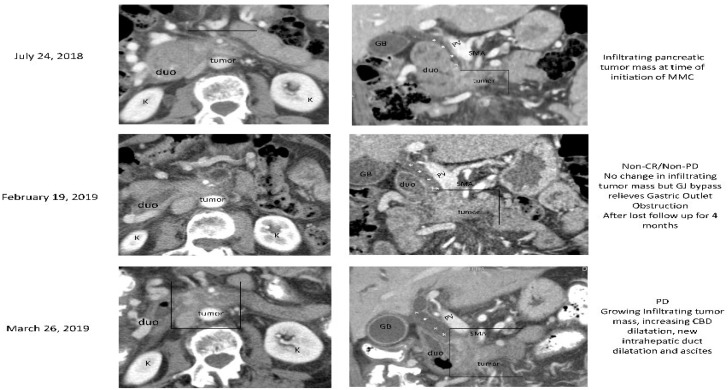
Case 2 treatment responses. Legend: Tumor response to Mitomcyin C from patient 2. Stable disease seen in primary pancreas mass (labelled ‘tumor’) between initiation of therapy at 24 July 2018 until 19 February 2019. Growth of primary tumor seen on 26 March 2019. Duo: duodenum, K: kidney, PV: portal vein, SMV: superior mesenteric vein, GB: gallbladder, SMA: superior mesenteric artery.

**Figure 4 biomedicines-10-02705-f004:**
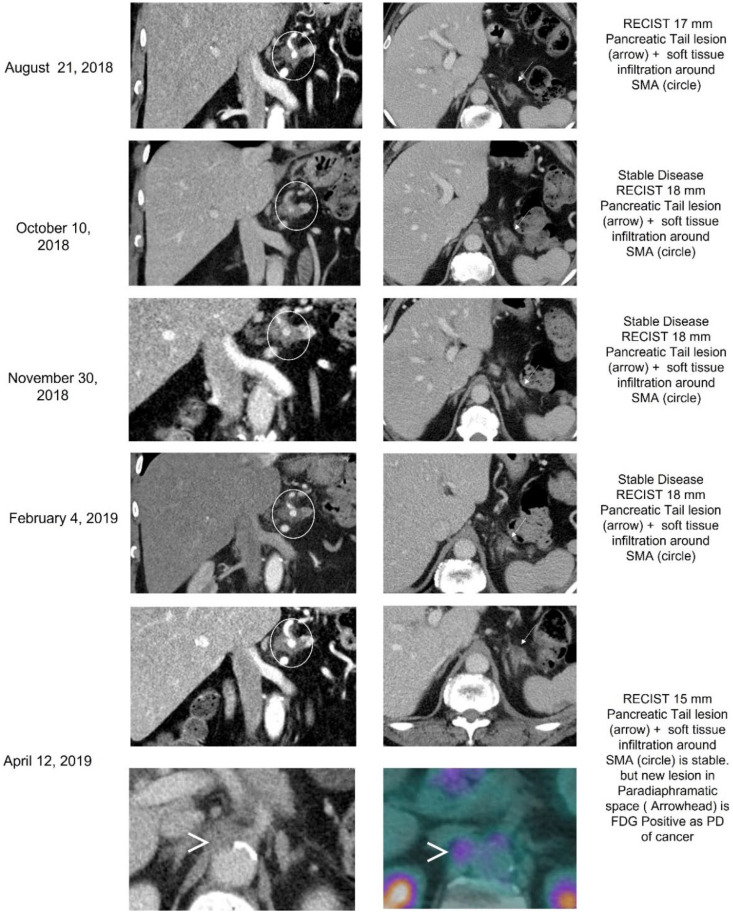
Case 3 treatment responses. Legend: Treatment with MMC resulted in stable disease in the pancreatic tail primary lesion (arrow). The soft tumor tissue around superior mesenteric artery was also stable (circle). On April 2019 disease progression was observed, with a new para diaphragmatic lesion, later was confirmed as a lung cancer (arrowhead).

**Figure 5 biomedicines-10-02705-f005:**
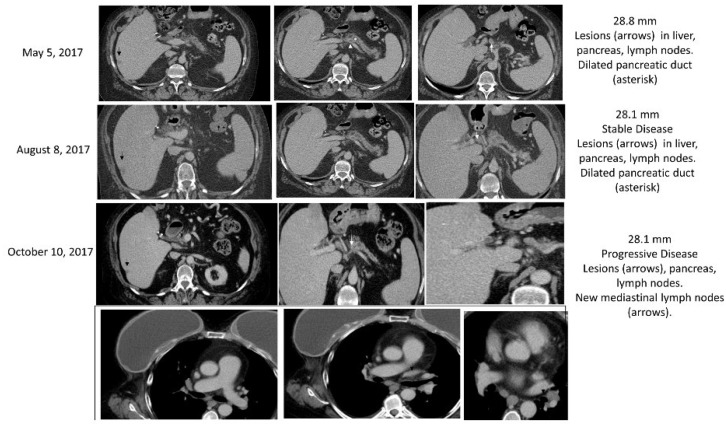
Case 4 treatment responses. Legend: Treatment with MMC resulted in stable disease, in multiple liver secondary lesions (arrows). Later, disease progression was observed with new mediastinal lymph nodes (images below).

**Figure 6 biomedicines-10-02705-f006:**
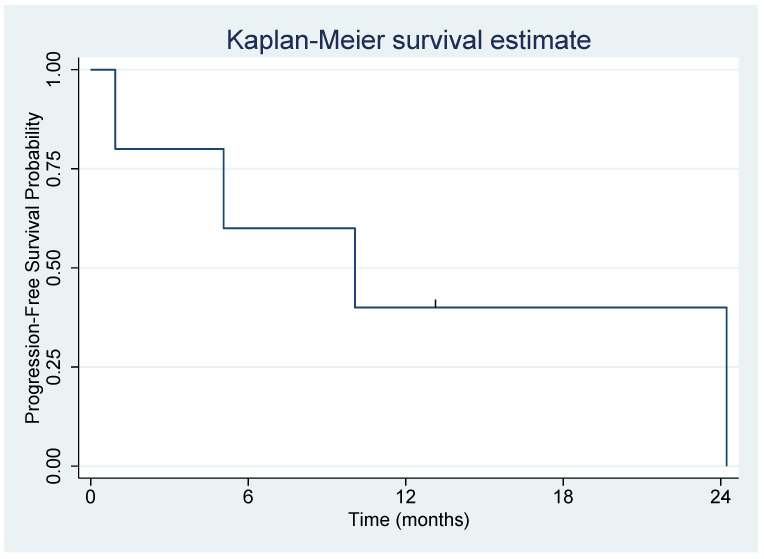
Overall median PFS on MMC treatment. Legend: Patient 3 did not progress and died of lung cancer, so was considered censored at the time of death. The median PFS was 10.1 months (95% CI = 0.9 months, upper bound not reached).

**Figure 7 biomedicines-10-02705-f007:**
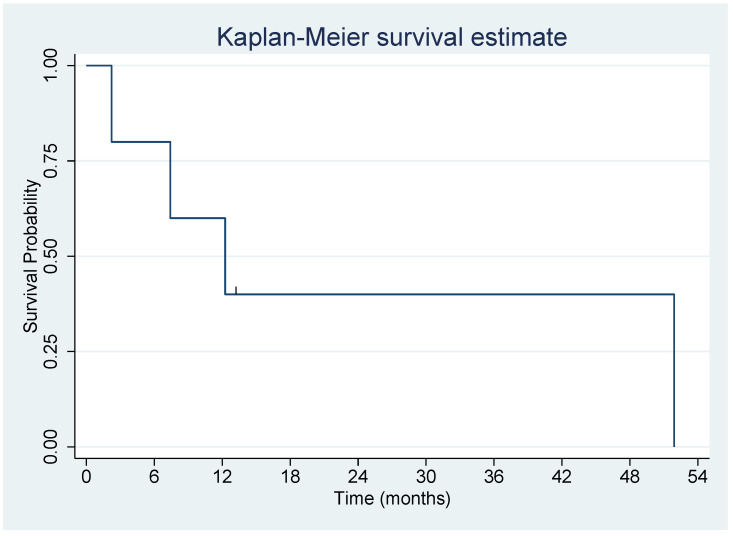
Overall survival on MMC treatment. Legend: Patient 3 was again censored at the time of death. The median OS was 12.3 months (95% CI = 2.2 months, upper bound not reached).

**Table 1 biomedicines-10-02705-t001:** Patient Demographic and Clinical Characteristics.

	Pancreatic Cancer (5 Patients)	
**Patient characteristics**	No.	%
**Age at diagnosis (years):**		
Median ± SD	58.3 ± 16.5	
Range	30–70	
Sex:		
Female	3	60
Male	2	40
**Site(s) of Metastatic Disease:**		
Liver	4	80
Lung/Bone	1	20
**No. of prior regimens for advanced disease prior to Mitomycin C therapy:**		
Mean	3.2	
SD	1.6	
**Personal history of prior malignancy:**		
Any malignancy	2	40
DNA repair-associated malignancy	1	20
**Family history of malignancy:**		
Any malignancy	4	80
DNA repair-associated malignancy	4	80
**Abbreviations: SD, standard deviation**		

**Table 2 biomedicines-10-02705-t002:** Patient’s genetic characteristics.

Patient #	Age of PC Diagnosis	Sex	Germline Mutation (s)	Somatic Mutation (s)	Personal Hx of Another Cancer	FPC	HBOC	Lynch
1	69	F	*BRCA2* c.3103G > T; p.E1035	*BRCA2* E1035, CDKN2A/2B deletion, KRAS Q61R, TP 53 S166, Low TMB 4, MSS	No	N	Y	N
2	61	F	*BRCA2*	MSS, ATM L1675, *BRCA2* Y3092fs 11, KRAS G12D, CDKN2A/B, MTAP loss exon 2-8, SMAD4 Q250	Breast	N	Y	N
3	70	M	*BRCA2* c.3362C > G (pathogenic); *BRCA2* c.2892A > T (VUS); MSH6 (c.-16C > A; POLD1 c.46A > G	*BRCA2* (pathogenic); *BRCA2* exon 11 p.K964N (VUS); MGMT pos, ARID1A pathogenic; KRAS mutated; ERCC1 pos; TOPO1 pos; TUBB3 pos	Lung	N	Y	N
4	58	F	*BRCA2, c.5946delT(p.Ser1982Argfs 22*	Not tested	No	N	Y	N
5	30	M	PALB2 c.1675_1676delinsTG (p.Gln559) (pathogenic); ATM c.2494C > T (p.Arg932Cys) (VUS); MLH1 c.1050A > G (silent) (VUS); CDK4 c.522 + 8G > A (intronic) (VUS); *BRCA1* c.5411T > A (p.Val1804Asp) (VUS)	ATM (P604S), TP53 (W146X) (Pathogenic), *BRCA1* mutation (V1804D) (VUS), low RRM1, PGP, TLE3, TUBB3, TOP2A	No	N	N	N

Legend: PC: Pancreatic cancer, VUS: variant of uncertain significance, FPC: familial pancreatic cancer, HBOC: hereditary breast and ovarian cancer.

**Table 3 biomedicines-10-02705-t003:** Patient’s outcome results.

Patient	Response	CA 19.9 Change (%)	PFS (Months)	OS (Months)
1	PR	↓ 68	24	51
2	SD	↓ 84	10	12
3	PR	↑ 220	13 #	13 #
4	SD	↓ 69	5	7
5	PD	↑ 49	1	2

Legend: # Patient 3 did not progress and died of lung cancer, so was considered censored at the time of death. SD: Stable disease, PR: Partial response, PD: progressive disease.

**Table 4 biomedicines-10-02705-t004:** Toxicities by CTCAE v 5.0 with MMC at initial dose of 8 mg/m^2^.

Patient	Nausea/Vomiting	Hematologic	Diarrhea
1	G2	Thrombocytopenia G2 Anemia G2	* G4
2	G2	Thrombocytopenia G3 Anemia G2	G2
3	G2	Thrombocytopenia G2 Anemia G2	None
4	G2	Thrombocytopenia G1 Anemia G3	None
5	G2	None	None

* Colitis.

## Data Availability

Not applicable.

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
