# Peer review of "Mitomycin C in Homologous Recombination Deficient Metastatic Pancreatic Cancer after Disease Progression on Platinum-Based Chemotherapy and Olaparib"

_biomedicines, 2022, doi:10.3390/biomedicines10112705_

Round 1

Reviewer 1 Report

The manuscript describes the treatment of homologous recombinant deficient (HRD) metastatic  pancreatic cancer (MPC)  with platinum-based chemotherapy and PARP  inhibitors. These results are  interesting , the manuscript is well organized; results and discussions are adequate. 

In specific

Introduction part of the manuscript need to improved with more citations.

Figure 1 seems to be a combination photograph, it showed be redrawn.

Author Response

Thank you for the review

Reviewer 1

The manuscript describes the treatment of homologous recombinant deficient (HRD) metastatic pancreatic cancer (MPC)  with platinum-based chemotherapy and PARP  inhibitors. These results are interesting, the manuscript is well organized; results and discussions are adequate. 

In specific

The introduction part of the manuscript needs to be improved with more citations.

R: Thank you for the suggestion, we have now included more references and backgrounds.

Figure 1 seems to be a combination photograph, it showed be redrawn.

R: Thank you for the suggestion, we have now redrawn Figure 1.

Reviewer 2 Report

Novel and encouraging data for MMC in pancreatic cancer

Author Response

Thank you for reviewing the manuscript, we improved the quality of the figures and references 

Reviewer 3 Report

In this article, the authors reported five HRD MPC patients treated with MMC were evaluated. All patients had tumors that progressed on platinum-based chemotherapy before having MMC treatment. The treatment with MMC seems to tolerate well on patients with germline BRCA ½ or PALB2 metastatic PC.

I have several comments as below:

1.       Fig 1: the authors should include figure legend and explain in detail what is presented in the graph. What does the horizon stand for? The graphs should be bigger. It is hard to know what the authors want to show in the graph.

2.       Fig 2: what is the difference between the left photo and the middle photo? The authors need a brief description for the Fig and a scale bar.

3.       Fig 3: the authors need a brief description for the Fig and a scale bar. Explain what “duo”, “k” etc. mean in the Fig legend.

4.       Fig 4: the authors need a brief description for the Fig and a scale bar. Why there is a rectangle around photos in April 12. What does it mean? Why do we have an arrow in the photo?

5.       Fig 5: the authors need a brief description for the Fig and a scale bar. There are 3 photos underneath Oct 10, 2017. What are they?

6.       Table 3: PR, SD stands for? It seems like patients 3 and 5 did not respond to MMS when the CA19.9 level went up. Why does patient 1 receive 3 courses mitomycin C while other patients receive only one course? Despite receiving 3 courses of mitomycin C, the CA19.9 change is only down 68% compared to patient 2 and 4. Do the authors know the reason why?

7.       Do the authors know the reason why patients fail the treatment with PARP inhibitors and platinum-based chemotherapy regimens but respond well with MMC? Only one patient shows this phenomenon. (Line 263-265).

Author Response

Thank you for comments, we have included our responses. 

Round 2

Reviewer 3 Report

The authors addressed all of my comments. The current version deserves to be published in Biomedicines journal.